# A New Convex Relaxation for Tensor Completion

**Bernardino Romera-Paredes**
Department of Computer Science
and UCL Interactive Centre
University College London
Malet Place, London WC1E 6BT, UK
B.RomeraParedes@cs.ucl.ac.uk

**Massimiliano Pontil**
Department of Computer Science and
Centre for Computational Statistics
and Machine Learning
University College London
Malet Place, London WC1E 6BT, UK
m.pontil@cs.ucl.ac.uk

## Abstract

We study the problem of learning a tensor from a set of linear measurements. A prominent methodology for this problem is based on a generalization of trace norm regularization, which has been used extensively for learning low rank matrices, to the tensor setting. In this paper, we highlight some limitations of this approach and propose an alternative convex relaxation on the Euclidean ball. We then describe a technique to solve the associated regularization problem, which builds upon the alternating direction method of multipliers. Experiments on one synthetic dataset and two real datasets indicate that the proposed method improves significantly over tensor trace norm regularization in terms of estimation error, while remaining computationally tractable.

## 1 Introduction

During the recent years, there has been a growing interest on the problem of learning a tensor from a set of linear measurements, such as a subset of its entries, see [9, 17, 22, 23, 25, 26, 27] and references therein. This methodology, which is also referred to as tensor completion, has been applied to various fields, ranging from collaborative filtering [15], to computer vision [17], and medical imaging [9], among others. In this paper, we propose a new method to tensor completion, which is based on a convex regularizer which encourages low rank tensors and develop an algorithm for solving the associated regularization problem.

Arguably the most widely used convex approach to tensor completion is based upon the extension of trace norm regularization [24] to that context. This involves computing the average of the trace norm of each matricization of the tensor [16]. A key insight behind using trace norm regularization for matrix completion is that this norm provides a tight convex relaxation of the rank of a matrix defined on the spectral unit ball [8]. Unfortunately, the extension of this methodology to the more general tensor setting presents some difficulties. In particular, we shall prove in this paper that the tensor trace norm is not a tight convex relaxation of the tensor rank.

The above negative result stems from the fact that the spectral norm, used to compute the convex relaxation for the trace norm, is not an invariant property of the matricization of a tensor. This observation leads us to take a different route and study afresh the convex relaxation of tensor rank on the Euclidean ball. We show that this relaxation is tighter than the tensor trace norm, and we describe a technique to solve the associated regularization problem. This method builds upon the alternating direction method of multipliers and a subgradient method to compute the proximity operator of the proposed regularizer. Furthermore, we present numerical experiments on one synthetic dataset and two real-life datasets, which indicate that the proposed method improves significantly over tensor trace norm regularization in terms of estimation error, while remaining computationally tractable.

The paper is organized in the following manner. In Section 2, we describe the tensor completion framework. In Section 3, we highlight some limitations of the tensor trace norm regularizer and present an alternative convex relaxation for the tensor rank. In Section 4, we describe a method to solve the associated regularization problem. In Section 5, we report on our numerical experience with the proposed method. Finally, in Section 6, we summarize the main contributions of this paper and discuss future directions of research.

## 2  Preliminaries

In this section, we begin by introducing some notation and then proceed to describe the learning problem. We denote by $\mathbb{N}$ the set of natural numbers and, for every $k \in \mathbb{N}$, we define $[k] = \{1, \dots, k\}$. Let $N \in \mathbb{N}$ and let[1] $p_1, \dots, p_N \geq 2$. An $N$-order tensor $\mathcal{W} \in \mathbb{R}^{p_1 \times \cdots \times p_N}$, is a collection of real numbers $(\mathcal{W}_{i_1,\dots,i_N} : i_n \in [p_n], n \in [N])$. Boldface Euler scripts, e.g. $\mathcal{W}$, will be used to denote tensors of order higher than two. Vectors are 1-order tensors and will be denoted by lower case letters, e.g. $x$ or $a$; matrices are 2-order tensors and will be denoted by upper case letters, e.g. $W$. If $x \in \mathbb{R}^d$ then for every $r \leq s \leq d$, we define $x_{r:s} := (x_i : r \leq i \leq s)$. We also use the notation $p_{\min} = \min\{p_1, \dots, p_N\}$ and $p_{\max} = \max\{p_1, \dots, p_N\}$.

A mode-$n$ fiber of a tensor $\mathcal{W}$ is a vector composed of the elements of $\mathcal{W}$ obtained by fixing all indices but one, corresponding to the $n$-th mode. This notion is a higher order analogue of columns (mode-1 fibers) and rows (mode-2 fibers) for matrices. The mode-$n$ matricization (or unfolding) of $\mathcal{W}$, denoted by $W_{(n)}$, is a matrix obtained by arranging the mode-$n$ fibers of $\mathcal{W}$ so that each of them is a column of $W_{(n)} \in \mathbb{R}^{p_n \times J_n}$, where $J_n := \prod_{k \neq n} p_k$. Note that the ordering of the columns is not important as long as it is used consistently.

We are now ready to describe the learning problem. We choose a linear operator $\mathcal{I} : \mathbb{R}^{p_1 \times \cdots \times p_N} \to \mathbb{R}^m$, representing a set of linear measurements obtained from a target tensor $\mathcal{W}^0$ as $y = \mathcal{I}(\mathcal{W}^0) + \xi$, where $\xi$ is some disturbance noise. Tensor completion is an important example of this setting, in this case the operator $\mathcal{I}$ returns the known elements of the tensor. That is, we have $\mathcal{I}(\mathcal{W}^0) = (\mathcal{W}^0_{i_1(j),\dots,i_N(j)} : j \in [m])$, where, for every $j \in [m]$ and $n \in [N]$, the index $i_n(j)$ is a prescribed integer in the set $[p_n]$. Our aim is to recover the tensor $\mathcal{W}^0$ from the data $(\mathcal{I}, y)$. To this end, we solve the regularization problem

$$\min \left\{ \|y - \mathcal{I}(\mathcal{W})\|_2^2 + \gamma R(\mathcal{W}) : \mathcal{W} \in \mathbb{R}^{p_1 \times \cdots \times p_N} \right\} \tag{1}$$

where $\gamma$ is a positive parameter which may be chosen by cross validation. The role of the regularizer $R$ is to encourage solutions $\mathcal{W}$ which have a simple structure in the sense that they involve a small number of "degrees of freedom". A natural choice is to consider the average of the rank of the tensor's matricizations. Specifically, we consider the combinatorial regularizer

$$R(\mathcal{W}) = \frac{1}{N} \sum_{n=1}^{N} \operatorname{rank}(W_{(n)}). \tag{2}$$

Finding a convex relaxation of this regularizer has been the subject of recent works [9, 17, 23]. They all agree to use the sum of nuclear norms as a convex proxy of $R$. This is defined as the average of the trace norm of each matricization of $\mathcal{W}$, that is,

$$\|\mathcal{W}\|_{\mathrm{tr}} = \frac{1}{N} \sum_{n=1}^{N} \|W_{(n)}\|_{\mathrm{tr}} \tag{3}$$

where $\|W_{(n)}\|_{\mathrm{tr}}$ is the trace (or nuclear) norm of matrix $W_{(n)}$, namely the $\ell_1$-norm of the vector of singular values of matrix $W_{(n)}$ (see, e.g. [14]). Note that in the particular case of 2-order tensors, functions (2) and (3) coincide with the usual notion of rank and trace norm of a matrix, respectively.

A rational behind the regularizer (3) is that the trace norm is the tightest convex lower bound to the rank of a matrix on the spectral unit ball, see [8, Thm. 1]. This lower bound is given by the convex envelope of the function

$$\Psi(W) = \begin{cases} \operatorname{rank}(W), & \text{if } \|W\|_\infty \leq 1 \\ +\infty, & \text{otherwise} \end{cases} \tag{4}$$

where $\|\cdot\|_\infty$ is the spectral norm, namely the largest singular value of $W$. The convex envelope can be derived by computing the double conjugate of $\Psi$. This is defined as

$$\Psi^{**}(W) = \sup\left\{\langle W, S\rangle - \Psi^*(W) : S \in \mathbb{R}^{p_1 \times p_2}\right\} \tag{5}$$

where $\Psi^*$ is the conjugate of $\Psi$, namely $\Psi^*(S) = \sup\left\{\langle W, S\rangle - \Psi(W) : W \in \mathbb{R}^{p_1 \times p_2}\right\}$.

Note that $\Psi$ is a spectral function, that is, $\Psi(W) = \psi(\sigma(W))$ where $\psi : \mathbb{R}^d_+ \to \mathbb{R}$ denotes the associated symmetric gauge function. Using von Neumann's trace theorem (see e.g. [14]) it is easily seen that $\Psi^*(S)$ is also a spectral function. That is, $\Psi^*(S) = \psi^*(\sigma(S))$, where

$$\psi^*(\sigma) = \sup\left\{\langle \sigma, w\rangle - \psi(w) : w \in \mathbb{R}^d_+\right\}, \quad \text{with } d := \min(p_1, p_2).$$

We refer to [8] for a detailed discussion of these ideas. We will use this equivalence between spectral and gauge functions repeatedly in the paper.

# 3 Alternative Convex Relaxation

In this section, we show that the tensor trace norm is not a tight convex relaxation of the tensor rank $R$ in equation (2). We then propose an alternative convex relaxation for this function.

Note that due to the composite nature of the function $R$, computing its convex envelope is a challenging task and one needs to resort to approximations. In [22], the authors note that the tensor trace norm $\|\cdot\|_{\text{tr}}$ in equation (3) is a convex lower bound to $R$ on the set

$$\mathcal{G}_\infty := \left\{\mathcal{W} \in \mathbb{R}^{p_1 \times \cdots \times p_N} : \left\|W_{(n)}\right\|_\infty \leq 1, \ \forall n \in [N]\right\}.$$

The key insight behind this observation is summarized in Lemma 4, which we report in Appendix A. However, the authors of [22] leave open the question of whether the tensor trace norm is the convex envelope of $R$ on the set $\mathcal{G}_\infty$. In the following, we will prove that this question has a negative answer by showing that there exists a convex function $\Omega \neq \|\cdot\|_{\text{tr}}$ which underestimates the function $R$ on $\mathcal{G}_\infty$ and such that for some tensor $\mathcal{W} \in \mathcal{G}_\infty$ it holds that $\Omega(\mathcal{W}) > \|\mathcal{W}\|_{\text{tr}}$.

To describe our observation we introduce the set

$$\mathcal{G}_2 := \left\{\mathcal{W} \in \mathbb{R}^{p_1 \times \cdots \times p_N} : \|\mathcal{W}\|_2 \leq 1\right\}$$

where $\|\cdot\|_2$ is the Euclidean norm for tensors, that is,

$$\|\mathcal{W}\|_2^2 := \sum_{i_1=1}^{p_1} \cdots \sum_{i_N=1}^{p_N} (\mathcal{W}_{i_1,\ldots,i_N})^2.$$

We will choose

$$\Omega(\mathcal{W}) = \Omega_\alpha(\mathcal{W}) := \frac{1}{N} \sum_{n=1}^{N} \omega_\alpha^{**}\left(\sigma\left(W_{(n)}\right)\right) \tag{6}$$

where $\omega_\alpha^{**}$ is the convex envelope of the cardinality of a vector on the $\ell_2$-ball of radius $\alpha$ and we will choose $\alpha = \sqrt{p_{\min}}$. Note, by Lemma 4 stated in Appendix A, that for every $\alpha > 0$, function $\Omega_\alpha$ is a convex lower bound of function $R$ on the set $\alpha\mathcal{G}_2$.

Below, for every vector $s \in \mathbb{R}^d$ we denote by $s^\downarrow$ the vector obtained by reordering the components of $s$ so that they are non increasing in absolute value, that is, $|s_1^\downarrow| \geq \cdots \geq |s_d^\downarrow|$.

**Lemma 1.** *Let $\omega_\alpha^{**}$ be the convex envelope of the cardinality on the $\ell_2$-ball of radius $\alpha$. Then, for every $x \in \mathbb{R}^d$ such that $\|x\|_2 = \alpha$, it holds that $\omega_\alpha^{**}(x) = \text{card}(x)$.*

This lemma is proved in Appendix B. The function $\omega_\alpha^{**}$ resembles the norm developed in [1], which corresponds to the convex envelope of the indicator function of the cardinality of a vector in the $\ell_2$ ball. The extension of its application to tensors is not straighforward though, as it is required to specify beforehand the rank of each matricization.

The next lemma provides, together with Lemma 1, a sufficient condition for the existence of a tensor $\mathcal{W} \in \mathcal{G}_\infty$ at which the regularizer in equation (6) is strictly larger than the tensor trace norm.

**Lemma 2.** *If $N \geq 3$ and $p_1, \ldots, p_N$ are not all equal to each other, then there exists $\mathcal{W} \in \mathbb{R}^{p_1 \times \cdots \times p_N}$ such that:* (a) $\|\mathcal{W}\|_2 = \sqrt{p_{\min}}$, (b) $\mathcal{W} \in \mathcal{G}_\infty$, (c) $\min_{n \in [N]} \mathrm{rank}(W_{(n)}) < \max_{n \in [N]} \mathrm{rank}(W_{(n)})$.

The proof of this lemma is presented in Appendix C. We are now ready to formulate the main result of this section.

**Proposition 3.** *Let $p_1, \ldots, p_N \in \mathbb{N}$, let $\| \cdot \|_{\mathrm{tr}}$ be the tensor trace norm in equation* (3) *and let $\Omega_\alpha$ be the function in equation* (6) *for $\alpha = \sqrt{p_{\min}}$. If $p_{\min} < p_{\max}$, then there are infinitely many tensors $\mathcal{W} \in \mathcal{G}_\infty$ such that $\Omega_\alpha(\mathcal{W}) > \|\mathcal{W}\|_{\mathrm{tr}}$. Moreover, for every $\mathcal{W} \in \mathcal{G}_2$, it holds that $\Omega_1(\mathcal{W}) \geq \|\mathcal{W}\|_{\mathrm{tr}}$.*

*Proof.* By construction $\Omega_\alpha(\mathcal{W}) \leq R(\mathcal{W})$ for every $\mathcal{W} \in \alpha \mathcal{G}_2$. Since $\mathcal{G}_\infty \subset \alpha \mathcal{G}_2$ then $\Omega_\alpha$ is a convex lower bound for the tensor rank $R$ on the set $\mathcal{G}_\infty$ as well. The first claim now follows by Lemmas 1 and 2. Indeed, all tensors obtained following the process described in the proof of Lemma 2 (in Appendix C) have the property that

$$
\begin{aligned}
\|\mathcal{W}\|_{\mathrm{tr}} &= \frac{1}{N} \sum_{n=1}^{N} \|\sigma(W_{(n)})\|_1 = \frac{1}{N} \left( p_{\min}(N-1) + \sqrt{p_{\min}^2 + p_{\min}} \right) \\
&< \frac{1}{N} \left( p_{\min}(N-1) + p_{\min} + 1 \right) = \Omega(\mathcal{W}) = R(\mathcal{W}).
\end{aligned}
$$

Furthermore there are infinitely many such tensors which satisfy this claim (see Appendix C). With respect to the second claim, given that $\omega_1^{**}$ is the convex envelope of the cardinality card on the Euclidean unit ball, then $\omega_1^{**}(\sigma) \geq \|\sigma\|_1$ for every vector $\sigma$ such that $\|\sigma\|_2 \leq 1$. Consequently,

$$
\Omega_1(\mathcal{W}) = \frac{1}{N} \sum_{n=1}^{N} \omega_1^{**} \left( \sigma\left(W_{(n)}\right) \right) \geq \frac{1}{N} \sum_{n=1}^{N} \|\sigma(W_{(n)})\|_1 = \|\mathcal{W}\|_{\mathrm{tr}}.
$$

$\square$

The above result stems from the fact that the spectral norm is not an invariant property of the matricization of a tensor, whereas the Euclidean (Frobenius) norm is. This observation leads us to further study the function $\Omega_\alpha$.

## 4 Optimization Method

In this section, we explain how to solve the regularization problem associated with the regularizer (6). For this purpose, we first recall the alternating direction method of multipliers (ADMM) [4], which was conveniently applied to tensor trace norm regularization in [9, 22].

### 4.1 Alternating Direction Method of Multipliers (ADMM)

To explain ADMM we consider a more general problem comprising both tensor trace norm regularization and the regularizer we propose,

$$
\min_{\mathcal{W}} \left\{ E\left(\mathcal{W}\right) + \gamma \sum_{n=1}^{N} \Psi\left(W_{(n)}\right) \right\} \tag{7}
$$

where $E(\mathcal{W})$ is an error term such as $\|y - \mathcal{I}(\mathcal{W})\|_2^2$ and $\Psi$ is a convex spectral function. It is defined, for every matrix $A$, as

$$
\Psi(A) = \psi(\sigma(A))
$$

where $\psi$ is a gauge function, namely a function which is symmetric and invariant under permutations. In particular, if $\psi$ is the $\ell_1$ norm then problem (7) corresponds to tensor trace norm regularization, whereas if $\psi = \omega_\alpha^{**}$ it implements the proposed regularizer.

Problem (7) poses some difficulties because the terms under the summation are interdependent, due to the different matricizations of $\mathcal{W}$ having the same elements rearranged in a different way. In

order to overcome this difficulty, the authors of [9, 22] proposed to use ADMM as a natural way to decouple the regularization term appearing in problem (7). This strategy is based on the introduction of $N$ auxiliary tensors, $\mathcal{B}_1, \ldots, \mathcal{B}_N \in \mathbb{R}^{p_1 \times \cdots \times p_N}$, so that problem (7) can be reformulated as[2]

$$\min_{\mathcal{W}, \mathcal{B}_1, \ldots, \mathcal{B}_N} \left\{ \frac{1}{\gamma} E\left(\mathcal{W}\right) + \sum_{n=1}^{N} \Psi\left(B_{n(n)}\right) \ : \ \mathcal{B}_n = \mathcal{W}, \ n \in [N] \right\} \tag{8}$$

The corresponding augmented Lagrangian (see e.g. [4, 5]) is given by

$$\mathcal{L}\left(\mathcal{W}, \mathcal{B}, \mathcal{A}\right) = \frac{1}{\gamma} E\left(\mathcal{W}\right) + \sum_{n=1}^{N} \left( \Psi\left(B_{n(n)}\right) - \langle \mathcal{A}_n, \mathcal{W} - \mathcal{B}_n \rangle + \frac{\beta}{2} \|\mathcal{W} - \mathcal{B}_n\|_2^2 \right), \tag{9}$$

where $\langle \cdot, \cdot \rangle$ denotes the scalar product between tensors, $\beta$ is a positive parameter and $\mathcal{A}_1, \ldots \mathcal{A}_N \in \mathbb{R}^{p_1 \times \cdots \times p_N}$ are the set of Lagrange multipliers associated with the constraints in problem (8).

ADMM is based on the following iterative scheme

$$\mathcal{W}^{[i+1]} \quad \leftarrow \quad \operatorname*{argmin}_{\mathcal{W}} \mathcal{L}\left(\mathcal{W}, \mathcal{B}^{[i]}, \mathcal{A}^{[i]}\right) \tag{10}$$

$$\mathcal{B}_n^{[i+1]} \quad \leftarrow \quad \operatorname*{argmin}_{\mathcal{B}_n} \mathcal{L}\left(\mathcal{W}^{[i+1]}, \mathcal{B}, \mathcal{A}^{[i]}\right) \tag{11}$$

$$\mathcal{A}_n^{[i+1]} \quad \leftarrow \quad \mathcal{A}_n^{[i]} - \left(\beta \mathcal{W}^{[i+1]} - \mathcal{B}_n^{[i+1]}\right). \tag{12}$$

Step (12) is straightforward, whereas step (10) is described in [9]. Here we focus on the step (11) since this is the only problem which involves function $\Psi$. We restate it with more explanatory notations as

$$\operatorname*{argmin}_{B_{n(n)}} \left\{ \Psi\left(B_{n(n)}\right) - \langle A_{n(n)}, W_{(n)} - B_{n(n)} \rangle + \frac{\beta}{2} \left\|W_{(n)} - B_{n(n)}\right\|_2^2 \right\}.$$

By completing the square in the right hand side, the solution of this problem is given by

$$\hat{B}_{n(n)} = \operatorname{prox}_{\frac{1}{\beta}\Psi}\left(X\right) := \operatorname*{argmin}_{B_{n(n)}} \left\{ \frac{1}{\beta} \Psi\left(B_{n(n)}\right) + \frac{1}{2} \left\|B_{n(n)} - X\right\|_2^2 \right\},$$

where $X = W_{(n)} - \frac{1}{\beta} A_{n(n)}$. By using properties of proximity operators (see e.g. [2, Prop. 3.1]) we know that if $\psi$ is a gauge function then

$$\operatorname{prox}_{\frac{1}{\beta}\Psi}\left(X\right) = U_X \operatorname{diag}\left(\operatorname{prox}_{\frac{1}{\beta}\psi}\left(\sigma(\mathrm{X})\right)\right) V_X^\top,$$

where $U_X$ and $V_X$ are the orthogonal matrices formed by the left and right singular vectors of $X$, respectively. If we choose $\psi = \|\cdot\|_1$ the associated proximity operator is the well-known soft thresholding operator, that is, $\operatorname{prox}_{\frac{1}{\beta}\|\cdot\|_1}\left(\sigma\right) = v$, where the vector $v$ has components

$$v_i = \operatorname{sign}\left(\sigma_i\right)\left(|\sigma_i| - \frac{1}{\beta}\right).$$

On the other hand, if we choose $\psi = \omega_\alpha^{**}$, we need to compute $\operatorname{prox}_{\frac{1}{\beta}\omega_\alpha^{**}}$. In the next section, we describe a method to accomplish this task.

## 4.2   Computation of the Proximity Operator

To compute the proximity operator of the function $\frac{1}{\beta}\omega_\alpha^{**}$ we will use several properties of proximity calculus. First, we use the formula (see e.g. [7]) $\operatorname{prox}_{g^*}\left(x\right) = x - \operatorname{prox}_g\left(x\right)$ for $g^* = \frac{1}{\beta}\omega_\alpha^{**}$. Next we use a property of conjugate functions from [21, 13], which states that $g(\cdot) = \frac{1}{\beta}\omega_\alpha^*(\beta\cdot)$. Finally, by the scaling property of proximity operators [7], we have that $\operatorname{prox}_g\left(x\right) = \frac{1}{\beta}\operatorname{prox}_{\beta\omega_\alpha^*}\left(\beta x\right)$.

**Algorithm 1** Computation of $\mathrm{prox}_{\beta\omega_\alpha^*}(y)$

---

**Input**: $y \in \mathbb{R}^d$, $\alpha, \beta > 0$.
**Output**: $\hat{w} \in \mathbb{R}^d$.
**Initialization**: initial step $\tau_0 = \frac{1}{2}$, initial and best found solution $w^0 = \hat{w} = P_S(y) \in \mathbb{R}^d$.
**for** $t = 1, 2, \ldots$ **do**
  $\tau \leftarrow \frac{\tau_0}{\sqrt{t}}$
  Find $k$ such that $k \in \mathrm{argmax}\left\{ \alpha\|w_{1:r}^{t-1}\|_2 - r : 0 \leq r \leq d \right\}$
  $\tilde{w}_{1:k} \leftarrow w_{1:k}^{t-1} - \tau\left( w_{1:k}^{t-1}\left(1 + \frac{\alpha\beta}{\|w_{1:k}^{t-1}\|_2}\right) - y_{1:k}\right)$
  $\tilde{w}_{k+1:d} \leftarrow w_{k+1:d}^{t-1} - \tau\left(w_{k+1:d}^{t-1} - y_{k+1:d}\right)$
  $w^t \leftarrow \tilde{P}_S(\tilde{w})$
  If $h(w^t) < h(\hat{w})$ then $\hat{w} \leftarrow w^t$
  If "Stopping Condition = True" then terminate.
**end for**

---

It remains to compute the proximity operator of a multiple of the function $\omega_\alpha^*$ in equation (13), that is, for any $\beta > 0$, $y \in S$, we wish to compute

$$\mathrm{prox}_{\beta\omega_\alpha^*}(y) = \underset{w}{\mathrm{argmin}}\left\{ h(w) : w \in S \right\}$$

where we have defined $S := \{w \in \mathbb{R}^d : w_1 \geq \cdots \geq w_d \geq 0\}$ and

$$h(w) = \frac{1}{2}\|w - y\|_2^2 + \beta \max_{r=0}^{d}\left\{\alpha\|w_{1:r}\|_2 - r\right\}.$$

In order to solve this problem we employ the projected subgradient method, see e.g. [6]. It consists in applying two steps at each iteration. First, it advances along a negative subgradient of the current solution; second, it projects the resultant point onto the feasible set $S$. In fact, according to [6], it is sufficient to compute an approximate projection, a step which we describe in Appendix D. To compute a subgradient of $h$ at $w$, we first find any integer $k$ such that $k \in \underset{r=0}{\overset{d}{\mathrm{argmax}}}\left\{\alpha\|w_{1:r}\|_2 - r\right\}$.

Then, we calculate a subgradient $g$ of the function $h$ at $w$ by the formula

$$g_i = \begin{cases} \left(1 + \frac{\alpha\beta}{\|w_{1:k}\|_2}\right)w_i - y_i, & \text{if } i \leq k, \\ w_i - y_i, & \text{otherwise.} \end{cases}$$

Now we have all the ingredients to apply the projected subgradient method, which is summarized in Algorithm 1. In our implementation we stop the algorithm when an update of $\hat{w}$ is not made for more than $10^2$ iterations.

## 5  Experiments

We have conducted a set of experiments to assess whether there is any advantage of using the proposed regularizer over the tensor trace norm for tensor completion[3]. First, we have designed a synthetic experiment to evaluate the performance of both approaches under controlled conditions. Then, we have tried both methods on two tensor completion real data problems. In all cases, we have used a validation procedure to tune the hyper-parameter $\gamma$, present in both approaches, among the values $\{10^j : j = -7, -6, \ldots, 1\}$. In our proposed approach there is one further hyper-parameter, $\alpha$, to be specified. It should take the value of the Euclidean norm of the underlying tensor. Since this is unknown, we propose to use the estimate

$$\hat{\alpha} = \sqrt{\|w\|_2^2 + (\mathrm{mean}(w)^2 + \mathrm{var}(w))\left(\prod_{i=1}^{N}p_i - m\right)},$$

where $m$ is the number of known entries and $w \in \mathbb{R}^m$ contains their values. This estimator assumes that each value in the tensor is sampled from $\mathcal{N}(\mathrm{mean}(w), \mathrm{var}(w))$, where $\mathrm{mean}(w)$ and $\mathrm{var}(w)$ are the average and the variance of the elements in $w$.

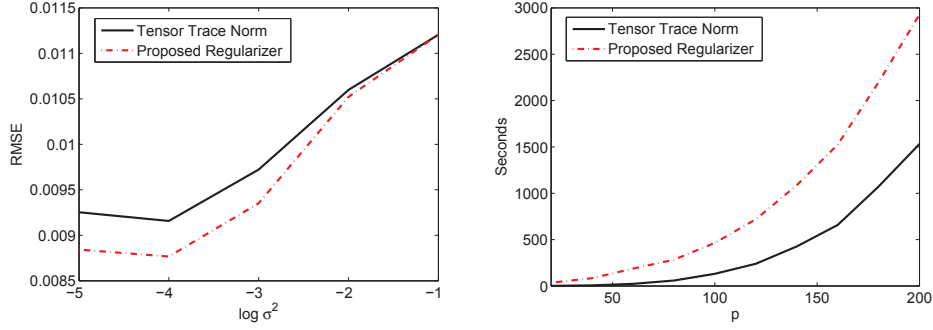

Figure 1: Synthetic dataset: (Left) Root Mean Squared Error (RMSE) of tensor trace norm and the proposed regularizer. (Right) Running time execution for different sizes of the tensor.

## 5.1  Synthetic Dataset

We have generated a 3-order tensor $\mathcal{W}^0 \in \mathbb{R}^{40 \times 20 \times 10}$ by the following procedure. First we generated a tensor $\mathcal{W}$ with ranks $(12, 6, 3)$ using Tucker decomposition (see e.g. [16])

$$\mathcal{W}_{i_1,i_2,i_3} = \sum_{j_1=1}^{12} \sum_{j_2=1}^{6} \sum_{j_3=1}^{3} \mathcal{C}_{j_1,j_2,j_3} M^{(1)}_{i_1,j_1} M^{(2)}_{i_2,j_2} M^{(3)}_{i_3,j_3}, \quad (i_1,i_2,i_3) \in [40] \times [20] \times [10]$$

where each entry of the Tucker decomposition components is sampled from the standard Gaussian distribution $\mathcal{N}(0, 1)$. We then created the ground truth tensor $\mathcal{W}^0$ by the equation

$$\mathcal{W}^0_{i_1,i_2,i_3} = \frac{\mathcal{W}_{i_1,i_2,i_3} - \mathrm{mean}(\mathcal{W})}{\sqrt{N}\mathrm{std}(\mathcal{W})} + \xi_{i_1,i_2,i_3}$$

where $\mathrm{mean}(\mathcal{W})$ and $\mathrm{std}(\mathcal{W})$ are the mean and standard deviation of the elements of $\mathcal{W}$, $N$ is the total number of elements of $\mathcal{W}$, and the $\xi_{i_1,i_2,i_3}$ are i.i.d. Gaussian random variables with zero mean and variance $\sigma^2$. We have randomly sampled $10\%$ of the elements of the tensor to compose the training set, $45\%$ for the validation set, and the remaining $45\%$ for the test set. After repeating this process 20 times, we report the average results in Figure 1 (Left). Having conducted a paired $t$-test for each value of $\sigma^2$, we conclude that the visible differences in the performances are highly significant, obtaining always $p$-values less than $0.01$ for $\sigma^2 \leq 10^{-2}$.

Furthermore, we have conducted an experiment to test the running time of both approaches. We have generated tensors $\mathcal{W}^0 \in \mathbb{R}^{p \times p \times p}$ for different values of $p \in \{20, 40, \ldots, 200\}$, following the same procedure as outlined above. The results are reported in Figure 1 (Right). For low values of $p$, the ratio between the running time of our approach and that of the trace norm regularization method is quite high. For example in the lowest value tried for $p$ in this experiment, $p = 20$, this ratio is $22.661$. However, as the volume of the tensor increases, the ratio quickly decreases. For example, for $p = 200$, the running time ratio is $1.9113$. These outcomes are expected because when $p$ is low, the most demanding routine in our method is the one described in Algorithm 1, where each iteration is of order $O(p)$ and $O(p^2)$ in the best and worst case, respectively. However, as $p$ increases the singular value decomposition routine, which is common to both methods, becomes the most demanding because it has a time complexity $O(p^3)$ [10]. Therefore, we can conclude that even though our approach is slower than the trace norm based method, this difference becomes much smaller as the size of the tensor increases.

## 5.2  School Dataset

The first real dataset we have tried is the Inner London Education Authority (ILEA) dataset. It is composed of examination marks ranging from 0 to 70, of 15362 students who are described by a set of attributes such as school and ethnic group. Most of these attributes are categorical, thereby we can think of exam mark prediction as a tensor completion problem where each of the modes corresponds to a categorical attribute. In particular, we have used the following attributes: school (139), gender (2), VR-band (3), ethnic (11), and year (3), leading to a 5-order tensor $\mathcal{W} \in \mathbb{R}^{139 \times 2 \times 3 \times 11 \times 3}$.

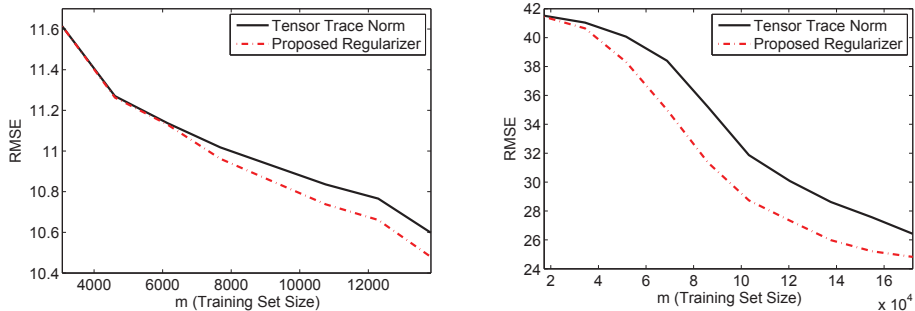

Figure 2: Root Mean Squared Error (RMSE) of tensor trace norm and the proposed regularizer for ILEA dataset (Left) and Ocean video (Right).

We have selected randomly $5\%$ of the instances to make the test set and another $5\%$ of the instances for the validation set. From the remaining instances, we have randomly chosen $m$ of them for several values of $m$. This procedure has been repeated 20 times and the average performance is presented in Figure 2 (Left). There is a distinguishable improvement of our approach with respect to tensor trace norm regularization for values of $m > 7000$. To check whether this gap is significant, we have conducted a set of paired $t$-tests in this regime. In all these cases we obtained a $p$-value below $0.01$.

## 5.3 Video Completion

In the second real-data experiment we have performed a video completion test. Any video can be treated as a 4-order tensor: "width" $\times$ "height" $\times$ "RGB" $\times$ "video length", so we can use tensor completion algorithms to rebuild a video from a few inputs, a procedure that can be useful for compression purposes. In our case, we have used the Ocean video, available at [17]. This video sequence can be treated as a tensor $\mathcal{W} \in \mathbb{R}^{160 \times 112 \times 3 \times 32}$. We have randomly sampled $m$ tensors elements as training data, $5\%$ of them as validation data, and the remaining ones composed the test set. After repeating this procedure 10 times, we present the average results in Figure 2 (Right). The proposed approach is noticeably better than the tensor trace norm in this experiment. This apparent outcome is strongly supported by the paired $t$-tests which we run for each value of $m$, obtaining always $p$-values below $0.01$, and for the cases $m > 5 \times 10^4$, we obtained $p$-values below $10^{-6}$.

## 6 Conclusion

In this paper, we proposed a convex relaxation for the average of the rank of the matricizations of a tensor. We compared this relaxation to a commonly used convex relaxation used in the context of tensor completion, which is based on the trace norm. We proved that this second relaxation is not tight and argued that the proposed convex regularizer may be advantageous. Our numerical experience indicates that our method consistently improves in terms of estimation error over tensor trace norm regularization, while being computationally comparable on the range of problems we considered. In the future it would be interesting to study methods to speed up the computation of the proximity operator of our regularizer and investigate its utility in tensor learning problems beyond tensor completion such as multilinear multitask learning [20].

### Acknowledgements

We wish to thank Andreas Argyriou, Raphael Hauser, Charles Micchelli and Marco Signoretto for useful comments. A valuable contribution was made by one of the anonymous referees. Part of this work was supported by EPSRC Grant EP/H017178/1, EP/H027203/1 and Royal Society International Joint Project 2012/R2.

## Footnotes

[1] For simplicity we assume that $p_n \geq 2$ for every $n \in [N]$, otherwise we simply reduce the order of the tensor without loss of information.

[2]The somewhat cumbersome notation $B_{n(n)}$ denotes the mode-$n$ matricization of tensor $\mathcal{B}_n$, that is, $B_{n(n)} = (\mathcal{B}_n)_{(n)}$.

[3]The code is available at http://romera-paredes.com/code/tensor-completion

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
