[Supplementary Material · tensor-nips_v9_appendix.pdf]

## Appendix

In this appendix, we describe an auxiliary result, prove the two lemmas formulated, and present the main steps for the computation of an approximate projection.

## A  A Useful Lemma

**Lemma 4.** *Let $\mathcal{C}_1, \ldots, \mathcal{C}_N$ be convex subsets of a Euclidean space and let $\mathcal{D} = \bigcap_{n=1}^N \mathcal{C}_n \neq \emptyset$. Let $g : \prod_{n=1}^N \mathcal{C}_n \to \mathbb{R}$ and let $h : \mathcal{D} \to \mathbb{R}$ be the function defined, for every $x \in \mathcal{D}$, as $h(x) = g(x, \ldots, x)$. Then, for every $x \in \mathcal{D}$, it holds that*

$$h^{**}(x) \geq g^{**}(x_1, \ldots, x_N)\big|_{x_n = x, \, \forall n \in [N]} \ .$$

*Proof.* Since the restriction of $g$ on $\mathcal{D}^N \subseteq \prod_{n=1}^N \mathcal{C}_n$ equals to $h$, the convex envelope of $g$ when evaluated on the smaller set $\mathcal{D}^N$ cannot be larger than the convex envelope of $h$ on $\mathcal{D}$. □

Using this result it is immediately possible to derive a convex lower bound for the function $R$ in equation (2). Since the convex envelope of the rank function on the unit ball of the spectral norm is the trace norm, using Lemma 4 with $\mathcal{C}_n = \{\mathcal{W} : \|W_{(n)}\|_\infty \leq 1\}$ and

$$g(\mathcal{W}_1, \ldots, \mathcal{W}_N) = \frac{1}{N} \sum_{n=1}^N \mathrm{rank}((W_n)_{(n)}),$$

we conclude that the convex envelope of the function $R$ on the set $\mathcal{G}_\infty$ is bounded from below by $\frac{1}{N} \sum_{n=1}^N \|W_{(n)}\|_{\mathrm{tr}}$. Likewise the convex envelope of $R$ on the set $\alpha \mathcal{G}_2$ is lower bounded by the function $\Omega_\alpha$ in equation (6).

## B  Proof of Lemma 1

*Proof.* First, we note that the conjugate of the function $\mathrm{card}$ on the $\ell_2$ ball of radius $\alpha$ is given by the formula

$$\omega_\alpha^* (s) = \sup_{\|y\|_2 \leq \alpha} \{\langle s, y \rangle - \mathrm{card}\,(y)\} = \max_{r \in \{0, \ldots, d\}} \{\alpha \|s_{1:r}^\downarrow\|_2 - r\}. \tag{13}$$

Hence, by the definition of the double conjugate, we have, for every $s \in \mathbb{R}^d$ that

$$\omega_\alpha^{**} (x) \geq \langle s, x \rangle - \max_{r \in \{0, \ldots, d\}} \{\alpha \|s_{1:r}^\downarrow\|_2 - r\}.$$

In particular, if $s = kx$ for some $k > 0$ this inequality becomes

$$\omega_\alpha^{**}(x) \geq k\|x\|_2^2 - \max_{r \in \{0, \ldots, d\}} (\alpha k \|x_{1:r}^\downarrow\|_2 - r).$$

If $k$ is large enough, the maximum is attained at $r = \mathrm{card}(x)$. Consequently,

$$\omega_\alpha^{**}(x) \geq k\alpha^2 - k\alpha^2 + \mathrm{card}(x) = \mathrm{card}(x).$$

By the definition of the convex envelope, it also holds that $\omega_\alpha^{**}(x) \leq \mathrm{card}(x)$. The result follows. □

## C  Proof of Lemma 2

*Proof.* Without loss of generality we assume that $p_1 \leq \cdots \leq p_N$. By hypothesis $p_1 < p_N$. First we consider the special case

$$p_1 = \cdots = p_{N-1}, \text{ and } p_N = p_1 + 1. \tag{14}$$

We define a class of tensors $\mathcal{W}$ by choosing a singular value decomposition for their mode-$N$ matricization,

$$\mathcal{W}_{i_1,i_2,\ldots,i_N} = \sum_{k=1}^{p_N} \sigma_k u_{i_N}^k v_{i_1,\ldots,i_{N-1}}^k \tag{15}$$

where $\sigma_1 = \cdots = \sigma_{p_N} = \sqrt{p_1/(p_1+1)}$, the vectors $u^k \in \mathbb{R}^{p_N}, \forall k \in [p_N]$ are orthonormal and the vectors $v^k \in \mathbb{R}^{p_1 p_2 \cdots p_{N-1}}, \forall k \in [p_N]$ are orthonormal as well. Moreover, we choose $v^k$ as

$$v_{i_1,\ldots,i_{N-1}}^k = \begin{cases} 1 & \text{if } i_1 = \cdots = i_{N-1} = k, & k < p_N \\ \frac{1}{\sqrt{p_1}} & \text{if } i_2 = \cdots = i_{N-1} = \text{module}(i_1,p_1)+1, & k = p_N \\ 0 & \text{otherwise.} \end{cases} \tag{16}$$

By construction the matrix $W_{(N)}$ has rank equal to $p_N$ and Frobenius norm equal to $\sqrt{p_1}$. Thus properties (a) and (c) hold true. It remains to show that $\mathcal{W}$ satisfies property (b). To this end, we will show, for every $n \in [N]$ and every $x \in \mathbb{R}^{p_n}$, that

$$\|W_{(n)}^\top x\|_2 \le \|x\|_2.$$

The case $n = N$ is immediate. If $n = 1$ we have

$$\begin{aligned}
\|W_{(1)}^\top x\|_2^2 &= \sum_{i_2,\ldots,i_N} \left( \sum_k \sigma_k \sum_{i_1} u_{i_N}^k v_{i_1,\ldots,i_{N-1}}^k x_{i_1} \right)^2 \\
&= \sum_{i_2,\ldots,i_N} \sum_{k,\ell} \sum_{i_1,j_1} x_{i_1} x_{j_1} \sigma_k \sigma_\ell u_{i_N}^k u_{i_N}^\ell v_{i_1,i_2,\ldots,i_{N-1}}^k v_{j_1,i_2,\ldots,i_{N-1}}^\ell \\
&= \sum_k \sigma_k^2 \sum_{i_1,j_1} x_{i_1} x_{j_1} \left( \sum_{i_2,\ldots,i_{N-1}} v_{i_1,i_2,\ldots,i_{N-1}}^k v_{j_1,i_2,\ldots,i_{N-1}}^k \right) \\
&= \sum_k \sigma_k^2 x_k^2 + \frac{\sigma_{p_N}^2}{p_1} \sum_k x_k^2 = \|x\|_2^2
\end{aligned}$$

where we used $\sum_{i_N} u_{i_N}^k u_{i_N}^\ell = \delta_{k,\ell}$ in the third equality, equation (16) and a direct computation in the fourth equality, and the definition of $\sigma_k$ in the last equality.
All other cases, namely $n = 2, \ldots, N-1$, are conceptually identical, so we only discuss the case $n = 2$. We have

$$\begin{aligned}
\|W_{(2)}^\top x\|_2^2 &= \sum_{i_1,i_3,\ldots,i_N} \left( \sum_k \sigma_k \sum_{i_2} u_{i_N}^k v_{i_2,\ldots,i_{N-1}}^k x_{i_2} \right)^2 \\
&= \sum_{i_1,i_3,\ldots,i_N} \sum_{k,\ell} \sum_{i_2,j_2} x_{i_2} x_{j_2} \sigma_k \sigma_\ell u_{i_N}^k u_{i_N}^\ell v_{i_1,i_2,\ldots,i_{N-1}}^k v_{i_1,j_2,\ldots,i_{N-1}}^\ell \\
&= \sum_k \sigma_k^2 \sum_{i_2,j_2} \left( x_{i_2} x_{j_2} \sum_{i_1,i_3,\ldots,i_{N=1}} v_{i_1,i_2,\ldots,i_{N-1}}^k v_{i_1,j_2,\ldots,i_{N-1}}^k \right) \\
&= \sum_k \sigma_k^2 x_k^2 + \frac{\sigma_{p_N}^2}{p_1} \sum_k x_k^2 = \|x\|_2^2
\end{aligned}$$

where again we used $\sum_{i_N} u_{i_N}^k u_{i_N}^\ell = \delta_{k,\ell}$ in the third equality, equation (16) and a direct computation in the fourth equality, and the definition of $\sigma_k$ in the last equality.
Finally, if assumption (14) is not true we set $\mathcal{W}_{i_1,\ldots,i_N} = 0$ if $i_n \ge p_1 + 1$, for some $n \le N - 1$ or $i_N > p_1 + 1$. We then proceed as in the case $p_1 = \cdots = p_{N-1}$ and $p_N = p_1 + 1$.

Note that one can build infinitely many tensors following this process, since the left singular vectors can be arbitrarily chosen in equation (15). □

# D    Computation of an Approximated Projection

Here, we address the issue of computing an approximate Euclidean projection onto the set

$$\mathcal{S} = \{v \in \mathbb{R}^d : v_1 \geq \cdots \geq v_d \geq 0\}.$$

That is, for every $v$, we shall find a point $\tilde{P}_{\mathcal{S}}(v) \in \mathcal{S}$ such that

$$\left\|\tilde{P}_{\mathcal{S}}(v) - z\right\|_2 \leq \|v - z\|_2, \ \forall z \in \mathcal{S}. \tag{17}$$

As noted in [6], in order to build $\tilde{P}_{\mathcal{S}}$ such that this property holds true, it is useful to express the set of interest as the smallest one in a series of nested sets. In our problem, we can express $\mathcal{S}$ as

$$\mathcal{S} = \mathcal{S}_d \subseteq \mathcal{S}_{d-1} \subseteq \ldots \subseteq \mathcal{S}_1,$$

where $\mathcal{S}_i := \left\{v \in \mathbb{R}^d : v_1 \geq v_2 \geq \ldots \geq v_i, v \geq 0\right\}$. This property allows us to sequentially compute an approximate projection on the set $\mathcal{S}$ using the formula

$$\tilde{P}_{\mathcal{S}}(v) = P_{\mathcal{S}_d}\left(P_{\mathcal{S}_{d-1}} \cdots \left(P_{\mathcal{S}_1}(v)\right)\right) \tag{18}$$

where, for every close convex set $\mathcal{C}$, we let $P_{\mathcal{C}}$ be the associated projection operator. Indeed, following [6], we can argue by induction on $i$ that $\tilde{P}_{\mathcal{S}}(v)$ verifies condition (17). The base case is $\|P_{\mathcal{S}_1}(v) - z\|_2 = \|v - z\|_2$, which is obvious. Now, if for a given $1 \leq i \leq d-1$ it holds that

$$\|P_{\mathcal{S}_i}(\cdots P_{\mathcal{S}_1}(v)) - z\|_2 \leq \|v - z\|_2$$

then

$$\left\|P_{\mathcal{S}_{i+1}}\left(P_{\mathcal{S}_i}(\cdots P_{\mathcal{S}_1}(v))\right) - z\right\|_2 \leq \|P_{\mathcal{S}_i}(\cdots P_{\mathcal{S}_1}(v)) - z\|_2 \leq \|v - z\|_2,$$

since $z$ is also contained in $\mathcal{S}_{i+1}$.

Note that to evaluate the right hand side of equation (18) we do not require full knowledge of $P_{\mathcal{S}_i}$, we only need to compute $P_{\mathcal{S}_{i+1}}(v)$ for $v \in \mathcal{S}_i$. The next proposition describes a recursive formula to achieve this step.

**Proposition 5.** *For any $v \in \mathcal{S}_i$, we express its first $i$ elements as $v_{1:i} = \left[v_{1:i-j}, v_i \mathbf{1}^j\right]$, where the last $j \in [i]$ is the largest integer such that $v_{i-j+1} = v_{i-j+2} = \cdots = v_i$, and $\mathbf{1}^d \in \mathbb{R}^d$ denotes the vector containing $1$ in all its elements. It holds that*

$$P_{\mathcal{S}_{i+1}}(v) = \begin{cases} v & \text{if } v_i \geq v_{i+1} \\ \left[v_{1:i-j}, \left(v_i + \frac{v_{i+1}-v_i}{j+1}\right)\mathbf{1}^{j+1}, v_{i+2:d}\right] & \text{if } v_i < v_{i+1} \text{ and } v_{i-j} \geq v_i + \frac{v_{i+1}-v_i}{j+1} \\ P_{\mathcal{S}_{i+1}}\left(\left[v_{1:i-j}, v_{i-j}\mathbf{1}^j, v_{i+1} - (v_{i-j}-v_i)j, v_{i+2:d}\right]\right) & \text{otherwise,} \end{cases}$$

*Proof.* The first case is straightforward. In the following we prove the remaining two. In both cases it will be useful to recall that the projection operator $P_{\mathcal{C}}$ on any convex set $\mathcal{C}$ is characterized as

$$x = P_{\mathcal{C}}(y) \iff \langle y - x, z - x \rangle \leq 0, \ \forall z \in \mathcal{C}. \tag{19}$$

To prove the second case, we use property (19) and apply simple algebraic transformations to obtain, for all $z \in \mathcal{S}_{i+1}$, that

$$\langle v - P_{\mathcal{S}_{i+1}}(v), z - P_{\mathcal{S}_{i+1}}(v)\rangle = \frac{v_{i+1} - v_i}{j+1}\left(jz_{i+1} - \|z_{i-j+1:i}\|_1\right) \leq 0.$$

Finally we prove the third case. We want to show that if $x = P_{\mathcal{S}_{i+1}}(v)$ then

$$x = P_{\mathcal{S}_{i+1}}\left(\left[v_{1:i-j}, v_{i-j}\mathbf{1}^j, v_{i+1} - (v_{i-j}-v_i)j, v_{i+2:d}\right]\right).$$

**Algorithm 2** Computing an approximated projection onto the set $\mathcal{S} = \{v \in \mathbb{R}^d : v_1 \geq \cdots \geq v_d \geq 0\}$.

---

**Input**: $y \in \mathbb{R}_+^d$.
**Output**: $v \in \mathcal{S}$.
**Initialization**: $v \leftarrow y$.
**for** $i = 1, 2, \ldots, d$ **do**
    **while** $v_i < v_{i+1}$ **do**
        $j \leftarrow \mathrm{argmax}\{\ell : \ell \in [i], v_i = v_{i-\ell+1}\}$
        **if** $v_{i-j} \geq v_i + \frac{v_{i+1}-v_i}{j+1}$ **then**
$$v_{1:i+1} \leftarrow \left[ v_{1:i-j}, \left( v_i + \frac{v_{i+1}-v_i}{j+1} \right) \mathbf{1}^{j+1} \right]$$
        **else**
$$v_{1:i+1} \leftarrow \left[ v_{1:i-j}, \, v_{i-j}\mathbf{1}^j, \, v_{i+1} - (v_{i-j} - v_i)\, j \right]$$
        **end if**
    **end while**
**end for**

---

By using property (19), the last equation is equivalent to the statement that if

$$\langle v - x, \, z - x \rangle \leq 0, \; \forall z \in \mathcal{S}_{i+1} \;\; \text{then} \tag{20}$$

$$\left\langle \left[ v_{1:i-j}, \, v_{i-j}\mathbf{1}^j, \, v_{i+1} - (v_{i-j} - v_i)\, j, \, v_{i+2:d} \right] - x, \, z - x \right\rangle \leq 0, \; \forall z \in \mathcal{S}_{i+1}. \tag{21}$$

A way to show that it holds true is to prove that the term in the left hand side of (21) is upper bounded by the corresponding term in (20). That is, for every $z \in \mathcal{S}_{i+1}$, we want to show that

$$\left\langle \left[ v_{1:i-j}, \, v_{i-j}\mathbf{1}^j, \, v_{i+1} - (v_{i-j} - v_i)\, j, \, v_{i+2:d} \right] - v, \, z - x \right\rangle \leq 0.$$

A direct computation yields the equivalent inequality

$$(v_{i-j} - v_i) \left( j x_{i+1} - \|x_{i-j+1:i}\|_1 + \|z_{i-j+1:i}\|_1 - j z_{i+1} \right) \leq 0. \tag{22}$$

Since $x = P_{\mathcal{S}_{i+1}}(v)$, $v_{i-j+1} = v_{i-j+2} = \cdots = v_i$ and $v_{i+1} > v_i$, then $x_{i-j+1} = x_{i-j+2} = \cdots = x_{i+1}$. Consequently, the left hand side of inequality (22) is equivalent to

$$(v_{i-j} - v_i) \left( \|z_{i-j+1:i}\|_1 - j z_{i+1} \right) \leq 0.$$

Note that the first factor is negative and the second is positive because $z$ and $v$ are in $\mathcal{S}_{i+1}$. The result follows. $\qquad\square$

Algorithm 2 summarizes our method to compute the approximated projection operator onto the set $\mathcal{S}$, based on Proposition 5.