[Reviews · NeurIPS 2013]

Submitted by Assigned_Reviewer_5

This paper proposes a new convex lower approximation for the average rank-n of tensors, which is useful for learning low rank tensors from a set of linear measurements. It is shown to be tighter (at some locations) than the commonly used trace norms that are extended from the matrix case. Efficient optimization algorithms are also designed based on alternating direction method of multipliers, together with an efficient algorithm for computing the proximal operator. Experiments on one synthetic dataset and two real datasets show that compared with the conventional trace norm regularization, the proposed relaxation yields statistically lower RMSE for tensor completion, with a tolerable increase in the computational cost.

The idea that motivates the new norm is very interesting, especially the key property of invariance to matricization in different modes. The experimental results are also encouraging. I would like to recommend accepting this paper for publication.

I am wondering if the l_2 ball in Lemma 1 can be extended to other norm balls. Based on the proof, the following two conditions will be necessary:
a) the resulting f**_\alpha(\sigma(W_(n))) in Eq 6 is independent of n,
b) \alpha is large enough such that \Gcal_\infty \subseteq \alpha \Gcal_2.

Here \Gcal stands for \mathcal{G}.

Lemma 1 can be stated in this generalized setting as:

Define f_\alpha as the convex envelope of the cardinality of x, card (x), on the ball of ||.|| with radius \alpha. Then for any x \in R^d such that ||x|| = \alpha, it holds f**_\alpha = card(x).

The proof is almost the same as that in the paper. In equation (7), change ||s_{1:r}||_2 to ||s_{1:r}||_*, where ||.||_* is the dual norm of ||.||. Then pick s = k \alpha y, where y = argmax_{||z||_* = 1} < z, x >. So ||y||_* = 1, and < y, x > = ||x|| = \alpha by the definition of dual norm. Then when k is large enough we have f**_\alpha (x) >= < k \alpha y, x > - (\alpha ||s||_* - card(x)) = k \alpha^2 – \alpha * (k \alpha *1) + card(x) = card(x).

Note the proof doesn’t need the condition a) or b).

Second, Proposition 2 can be extended as:

There exists \Wcal \in \Gcal_\infty such that f_\alpha(\Wcal) > |||\Wcal|||_tr.
Furthermore, for every \Wcal in the unit ball of ||.||, it holds that F_1(\Wcal) > |||\Wcal|||_tr.

Indeed, the second clause is trivial by the definition of Fenchel biconjugation (it’s the max of all convex functions that minorize \Omega(\Wcal)). The first clause does need the above assumptions a) and b).

I suspect the l_2 norm used in Lemma 1 is the only norm that satisfies these two conditions. It will be interesting to prove or disprove it. But of course, this is just a sufficient but not necessary condition for there being a \Wcal such that the induced norm on \Wcal is greater than |||\Wcal|||_tr.

Minor comments:
1. Line 139: \Wcal \in \Gcal_2 => \Wcal \in \sqrt{p_min} \Gcal_2
2. In various places, ||\Wcal||_tr should be changed into |||\Wcal|||_tr.
3. ADM: isn’t Alternating Direction Method of Multipliers more commonly abbreviated as ADMM?
4. It may be useful to highlight that the proposed F_\alpha(\Wcal) is NOT uniformly greater than \Omega(\Wcal), ie it doesn’t dominate \Omega(\Wcal).
5. What’s the value of relative RMSE in experiment? That is, RMSE divided by the ground truth value.
Summary: This paper proposed an interesting convex relaxation to the rank of tensor, which is tighter than the conventional trace norm at some points. Efficient optimization algorithms are provided and the experimental results are encouraging.

Submitted by Assigned_Reviewer_6

This paper proposes a new convex relaxation for the tensor rank minimization problem. The new convex regularizer is proved to be tighter than the trace norm, which is defined as the average of the nuclear norms of the matricizations. The idea is interesting. The main problem is that computing the proximal mapping of the new regularizer requires an iterative subgradient method, which might take lots of iterations and thus computionally expensive. In the numerical part, the authors should discuss the effect if different alpha is chosen, and also report the cpu times for different problems.
Summary: The authors should discuss the effect if different alpha is chosen.

Submitted by Assigned_Reviewer_7

The paper is about tensor estimation under low-rank assumption. The authors claim that the trace-norm suggested by references [7,15,20] by Gandy et al. 11, Liu et al. 09 and Signoretto et al. does not provide a tight relaxation of the index of interest that is a generalization of the rank of matrices to tensors. The authors support their argument by a counter-example (page 4), a theoretical result (Lemma 1 page 3) and empirical results on both real and synthetic data. They suggest an ADMM (please write it with 2 "M"s as it is the convention to make clear that it is the same algorithm) scheme for optimization with the suggested penalty. To obtain the penalty they suggest a bi-conjugation which is the standard way to do it.

I am not sure that the convex relaxation should be done on the euclidean ball as the authors do it. In the case of the trace norm it is done on the operator or spectral norm ball. For the max norm it is done on the element-wise ell_infty norm ball then approximated by the max-norm. Why should the euclidean norm define the ball on which to relax the rank?
Minor question: what is the cost of coputing the prox (Algorithm 1)? each iteration has cost p^2, this is huge, what is the stopping criterion implying about the computational cost?
Summary: The paper suggests a convex relaxation of the tensor rank over the euclidean norm ball, which is intriguing but not convincing to me.

Submitted by Assigned_Reviewer_8

**** Updated after authors' feedback ****
My main concerns are still that
- the improvement compared to L1 (or trace norm) comes from the discontinuousness at the boundary of the domain, where the solution almost never lies (as the authors point out).
- on the other hand, probably the proposed regularizer agrees with the L1 if ||w||_1 ≤ alpha, because it agrees with L1 on all the points on the L2 ball that have cardinality one, and the convex envelope of those points defines the ball ||w||_1 ≤ alpha.
*****************************************


This paper proposes a new convex relaxation of the tensor multi-linear rank based on the convex envelope of the cardinality function intersected with an L2-norm ball of some radius alpha.

The proposed convex relaxation is in the line of previous studies [7,15,19,20,22] that studied trace norm of the unfoldings (matricizations) of a tensor along different modes. The current paper points out that the tensor trace norm used in previous studies is not the tightest convex relaxation with respect to the Frobenius norm (L2 norm of the spectrum). To fix this, the authors apply the convex envelope of the cardinality function intersected with an L2-norm ball to the spectra of the unfoldings of a given tensor.

The authors show that the proposed convex envelope (for cardinality) is tight for any vector having the prespecified radius alpha. The trace norm is the tightest convex relaxation of the rank in terms of the spectral norm. The authors argue however that the spectral norm is not invariant to the choice of the mode to unfold a given tensor, nor is the tightness of trace norm. On the other hand, the Frobenius norm is invariant, so is the tightness of the proposed convex relaxation.

Some numerical experiments show that the proposed convex relaxation outperforms tensor trace norm in tensor completion.

Strength:
- The proposed convex relaxation is new and it seems to work.
- The idea might be used for inducing sparsity in general.
Weakness:
- The result of Lemma 1 seems applicable to other sprasity problems (sparsity, group sparsity, low-rank matrices). The authors should clarify how much of the contribution is specific to low-rank tensors.
- The authors do not prove that the proposed regularizer is the tightest convex relaxation for the combinatorial regularizer Omega(W) (they only show the tightness of the convex envelope function used to construct the regularizer in Lemma 1 and it is *tighter* than the trace norm in Prop 2).
- The proposed convex relaxation seems rather hard to compute. It is infinity for ||x|| > alpha and it has many discontinuities for ||x||=alpha. This probably makes the optimization and also proving something about the estimator hard.
- It would be nice if the authors could discuss any connection between this work and the k-support norm proposed in Argyriou et al. NIPS 2012)


Minor comments:
- The statement "if F_alpha(W) ≤ Omega(W) for every W\in G_2 then F_alpha(W) ≤ Omega(W) for every G_infty." (line 138) seems false, because G_2 is a smaller set than G_infty.
- The choice of alpha as in line 321 may not be the best, because the proposed regularizer is infinity for ||W||_F > alpha. It should be chosen more carefully.
- It was not clear how tight the proposed regularizer is when ||W||_F < alpha. I guess it equals the trace norm when ||W||_tr ≤ alpha (Correct me if I am wrong).
- What kind of performance guarantee can you prove for this regularizer? How does that compare to Tomioka et al. "Statistical Performance of Convex Tensor Decomposition" (NIPS 2011)?
Summary: This paper proposes a new convex relaxation for tensor multi-linear rank. The new regularizer seems interesting and it seems to work but it was not clear how much it is specific to tensors. In addition, though convex the regularizer might be ill-behaving (not continuous).
Author Feedback

Author rebuttal: Rev 5
Your question about looking for norms other than l_2 which can generalize the content of our paper is very interesting and it requires more investigation. To summarize our selection of the l_2 ball we can state that:
All norms of a matrix that can be written as a vector norm of the matrix elements are clearly matricization invariant norms (e.g. element-wise l_p norm). Among them, the l_2 (Frobenius) norm is perhaps the only one which is also a spectral function, allowing us to use von Neumann's trace theorem to compute the convex envelope of the rank of a matrix.

We agree with the first four minor comments (in the 4th one, we will highlight that F_\alpha(\Wcal) \leq \Omega(\Wcal) for all \Wcal \in the l_2 ball of radius \alpha).
Regarding the last comment, these are the relative RMSE results:
Synthetic dataset:
Tr Norm Our Regularizer
0.8168 0.7867
0.8243 0.7890
0.8674 0.8347
0.9513 0.9445
1.0000 1.0000
School dataset:
Tr Norm Our Regularizer
0.6132 0.6050
0.5923 0.5829
0.5755 0.5642
0.5622 0.5520
0.5566 0.5472
0.5494 0.5410
0.5449 0.5367
0.5422 0.5342
0.5363 0.5293
Video completion:
Tr Norm Our Regularizer
0.3227 0.3226
0.3194 0.3169
0.3121 0.2998
0.2989 0.2717
0.2802 0.2446
0.2588 0.2245
0.2412 0.2176
0.2273 0.2135
0.2159 0.2105

Rev 6
We have performed one more experiment varying the value of alpha in the synthetic dataset (for sigma^2=10^-4 and the training set being composed of 10% of the tensor elements), where we know that the ground truth’s alpha=1. The results are as follows:
log_10(alpha) RMSE
-1 0.0111
-0.5 0.0110
-0.25 0.0099
-0.15 0.0091
-0.1 0.0090
-0.05 0.0089
0 0.0089
0.05 0.0089
0.1 0.0090
0.15 0.0092
0.25 0.0093
0.5 0.0093
1 0.0093
whereas trace norm’s RMSE = 0.0093. Note that our approach performs better than trace norm for values of alpha around the ground truth. It does not perform well when alpha is much smaller than the ground truth, whereas it obtains better or identical RMSE when the value of alpha is overestimated. Experiments on the School Dataset (resp. Video Completion) show similar trends, namely our approach performs better than the trace norm for values of alpha in the range [10^-0.5, 10^1] (resp. [10^-0.25, 10^0.25]) times the value of alpha in eq. line 321.

Note that in lines 361-372 and Fig. 1 right, we report experiments regarding the cpu time for different tensor sizes.

Rev 7
We employ the Frobenius norm because it is invariant wrt. different matricizations of a tensor, a property which is lacked by the spectral norm (see also answer to Rev. 1).
Regarding the minor question, the reason for each iteration of the prox being quadratic is due to the projection algorithm, which in the worst case (i.e. when the input is sorted in opposite order) is quadratic. In practice, we have noticed that the input to the projection algorithm is usually very far from the worst case - the intuition behind this is that the input is the result of advancing toward a negative subgradient from an already sorted vector.
Furthermore, we have performed one more experiment to compare the average (over 1000 trials) number of iterations (ANI) required by the prox algorithm to terminate (we stop the Algorithm 1 when an update to {\hat w} is not made for more than 250 consecutive iterations) for different values of p, which indicates that this number changes gracefully with p:
p ANI
20 340.96
40 386.00
80 451.61
160 472.87

Rev 8
Weaknesses:
- The result in Lemma 1 can indeed be applied to other sparsity problems and its study could be an interesting extension of this paper. Here, we focus on tensors, where the advantage of using the Frobenius norm over the spectral norm is clear, due to its invariance to the choice of the mode to unfold a given tensor.
- Because of the composite nature of the combinatorial regularizer Omega, it seems hopeless to find the tightest convex relaxation of Omega.
- The discontinuity on the boundary does not seem an issue for optimization. Indeed we verified that in our simulations the optimal solution found has always norm smaller than alpha.
- Thank you for bringing up to our attention the paper by Argyriou et al., which we will cite in the revised version. There are some important differences between our work and that paper: a) They consider the convex envelope of the *indicator function* of the cardinality of a vector on the L2 ball, we consider the convex envelope of the cardinality of a vector on the L2 ball; b) They entirely focus on vector problems, we consider tensor problems; c) perhaps most importantly, their regularizer does not seem easily applicable to tensors because it requires one to specify N hyperparameters (the bound on the rank of each matricization involved in the convex relaxation). This fact makes the adaptation of their regularizer computationally challenging in the tensor setting.

Minor comments:
- You are right, this should be modified to "F_alpha(W) <= Omega(W) for every W \in sqrt{p_min} G_2 for every..."
- The choice of alpha is based on a reasonable heuristic which seems to work well in practice (see also answer to Rev. 2). Choosing alpha more carefully (e.g. by cross validation) would potentially lead to further improvements in the performance, at the cost of additional computation.
- Since the level sets of F_alpha are closed, then there exists W s.t. ||W||_F < alpha and F_alpha(W) > |||W|||_tr. Indeed, f^**_alpha is lower semi continuous (l.s.c.), see Thm. 11.1 in Rockafellar and Wets. Consequently, the function F_alpha is l.s.c. as well (Thm. 1.39). In turn this implies that for every eps > 0 there exists a point W in the interior of the alpha ball s.t. F_alpha(W) = Omega(W)-eps.
- We have not made a statistical analysis of the method yet; possibly Rademacher bounds could be derived, but this is subject for future work.